# Review of Pharmacological Strategies with Repurposed Drugs for Hereditary Hemorrhagic Telangiectasia Related Bleeding

**DOI:** 10.3390/jcm9061766

**Published:** 2020-06-06

**Authors:** Virginia Albiñana, Angel M. Cuesta, Isabel de Rojas-P, Eunate Gallardo-Vara, Lucía Recio-Poveda, Carmelo Bernabéu, Luisa María Botella

**Affiliations:** 1Centro de Investigaciones Biológicas Margarita Salas, Consejo Superior de Investigaciones Científicas (CSIC), 9 Ramiro de Maeztu Street, 28040 Madrid, Spain; vir_albi_di@yahoo.es (V.A.); acme@cib.csic.es (A.M.C.); iderojas@ucm.es (I.d.R.-P.); lrecio@cib.csic.es (L.R.-P.); bernabeu.c@cib.csic.es (C.B.); 2Centro de Investigación Biomédica en Red de Enfermedades Raras (CIBERER), Institute of Health Carlos III, 28040 Madrid, Spain; 3Yale Cardiovascular Research Center, Section of Cardiovascular Medicine, Department of Internal Medicine, Yale University School of Medicine, 300 George Street, New Haven, CT 06511, USA; eunate.gallardo@yale.edu

**Keywords:** HHT, ALK1, endoglin, raloxifene, bazedoxifene, tranexamic acid, propranolol, FK506, etamsylate, N-acetylcysteine

## Abstract

The diagnosis of hereditary hemorrhagic telangiectasia (HHT) is based on the Curaçao criteria: epistaxis, telangiectases, arteriovenous malformations in internal organs, and family history. Genetically speaking, more than 90% of HHT patients show mutations in *ENG* or *ACVRL1/ALK1* genes, both belonging to the TGF-β/BMP9 signaling pathway. Despite clear knowledge of the symptoms and genes of the disease, we still lack a definite cure for HHT, having just palliative measures and pharmacological trials. Among the former, two strategies are: intervention at “ground zero” to minimize by iron and blood transfusions in order to counteract anemia. Among the later, along the last 15 years, three different strategies have been tested: (1) To favor coagulation with antifibrinolytic agents (tranexamic acid); (2) to increase transcription of *ENG* and *ALK1* with specific estrogen-receptor modulators (bazedoxifene or raloxifene), antioxidants (N-acetylcysteine, resveratrol), or immunosuppressants (tacrolimus); and (3) to impair the abnormal angiogenic process with antibodies (bevacizumab) or blocking drugs like etamsylate, and propranolol. This manuscript reviews the main strategies and sums up the clinical trials developed with drugs alleviating HHT.

## 1. Introduction

Hereditary hemorrhagic telangiectasia (HHT) or Rendu-Osler-Weber syndrome is a genetic dominant autosomal multisystemic vascular disease, whose penetrance increases with age. The Curaçao criteria were designed to diagnose HHT, and include its clinical symptoms which are spontaneous and recurrent epistaxis (nose bleeds), mucocutaneous telangiectases, visceral localization (gastrointestinal telangiectases and/or arteriovenous malformations (AVMs), mainly in lung, brain or liver), and a first degree family member with a definite diagnosis of HHT (Figure 1A) [1,2,3]. The prevalence of HHT varies between 1:5000 and 1:8000 on average, although because of the “founder effect” and “insulation effect,” the prevalence is higher in some regions such as the Jura region in France, Funen Island in Denmark and the Caribbean Dutch Antilles [3,4,5]. Heterozygous mutations in either *ENDOGLIN (ENG)* or *ACVRL1/ALK1* genes trigger the pathogenesis of HHT in over 90% of HHT patients [6,7]. Less common mutations, responsible for 2% of HHT cases, appear in the *SMAD4* gene, leading to a combined syndrome of Juvenile Polyposis HHT (JPHT) [8] consisting of HHT symptoms, colon polyps and thoracic aneurysms [9]. Furthermore, chromosomes 5 and 7 have been described to possess two *loci* with unknown genes, that cause HHT3 [10] and HHT4, respectively [11]. An HHT-like syndrome called HHT5 has been linked to mutations in *BMP9* [12]. All mutations leading to HHT are found in genes belonging to the family of BMP9/TGF-β signaling pathway (Figure 1B).

Moreover, the capillary malformation (CM)/AVM syndrome is phenotypically similar to HHT, and is characterized by the appearance of multiple CMs that are small and red, round to oval shaped with a peripheral white halo and randomly distributed. These are linked to heterozygous pathogenic variants in *EPHB4* or *RASA1* identified by molecular genetic testing [13].

This review will focus on the pharmacological treatment for bleeding in HHT patients. With 93% of patients suffering light to moderate bleedings, epistaxis presents as the most frequent clinical manifestation of HHT [14,15]. It affects over 90% of patients before the age of 21, normally interfering with their quality of life [16]. Epistaxis are due to the telangiectases of the nasal mucosa, focally dilated venules, often connected directly with dilated arterioles [17]. Directly related to epistaxis is gastrointestinal (GI) bleeding, because of telangiectases in the digestive tract and observed in up to 80% of HHT patients [18]. However, GI bleeding becomes more frequent with age [19]. Although currently there is no optimal available treatment for either epistaxis or GI bleeding, the systemic pharmacological treatments that are used for epistaxis might also be useful to manage GI bleedings.

The pharmaceutical therapies which that are discussed in the following sections address therapies wherein the disease is due to heterozygous germ-line mutations in all cells of the HHT patient. These therapies may not be effective for some cutaneous telangiectases, wherein endothelial cells (EC) may have homozygous mutants for *ALK1*/*ENG* according to a recent publication of Snellings et al. [20].

## 2. General Care and Control of Anemia

To prevent crusting and allow the nasal mucosa to be correctly hydrated in HHT patients, local moisturizing treatments such as humidification, nasal cleaning with a saline solution and lipid-based topical ointments are used [18]. Despite these options, it is challenging to completely avoid nasal or GI bleeding in HHT, often leading to iron deficiency and anemia in these patients. For this reason, the first line treatment of HHT is focused on managing the anemia resulting from bleeding. Iron-enriched diets and iron supplements are cost-effective steps that significantly reduce the need of blood transfusions although the latter may be necessary in severely affected patients [2,21].

## 3. Therapeutic Pathways/Strategies of Pharmacological Treatments for HHT

The following section focuses on reviewing the pharmacological treatments, from a preclinical perspective. Robert et al. have also recently reviewed this topic [22].

Options to control nose and GI bleeding could be used, according to the following strategies (Table 1). It should be commented that the drugs were included into each group, according to the main mechanism observed in vitro and/or in vivo experiments, yet in some cases other additional mechanisms maybe contributing to the therapeutic effect.

-**Strategy 1**. Although HHT does not result from a clotting failure, the use of antifibrinolytics to restore the balance between coagulation versus fibrinolysis would help to promote a quicker coagulation and to stabilize the fibrin network. Among the antifibrinolytics used for epistaxis treatment, tranexamic acid (TA) and ε-aminocaproic acid (AC) stand out [23,24].-**Strategy 2**. HHT is associated with haploinsufficiency in *ENG* or *ALK1* genes, therefore stimulating their protein expression is thought to revert the HHT phenotype. At this point, raloxifene hydrochloride and bazedoxifene acetate, two specific estrogen receptor modulators (SERMs), have proven efficiency and safety, and have been designated as orphan drugs for HHT (2010 EU/3/10/730 and 2014 EU/3/14/1367; respectively) [25,26].-**Strategy 3**. Antiangiogenic therapies tackle the excess of abnormal vasculature present on the nasal mucosa in HHT. Therefore, bevacizumab (BZ) (Avastin^®^), a humanized monoclonal antibody against the main angiogenic factor, the vascular endothelial growth factor (VEGF), has been widely used and tested on HHT. Its systemic administration has improved hepatic function, delaying the liver transplant [27] but it has not shown consistent results when tested to decrease epistaxis events by topical spray administration [28,29].

Following the same antiangiogenic strategy, pazopanib, thalidomide, and more recently pomalidomide, have been used to inhibit the VEGF pathway.

Similarly, other cardiovascular drugs such as propranolol and timolol (non-specific β-blockers) have shown their benefits in nose bleeding when administered topically (both) and systemically (propranolol) [30] in HHT patients. Recently, the use of the fibroblast growth factor receptor (FGFR) blocker etamsylate, by local spray administration has been proven effective and has been designated as orphan drug for HHT in 2018 (EU/3/18/2087) [31].

Table 2 and Table 3, respectively, summarize the ongoing and finished clinical trials, conducted in HHT with the different drugs, most of them mentioned in this review. In addition, some other recent candidates like Vitamin D, itraconazole and doxycyclin are in ongoing clinical trials.

All the above-mentioned drugs used were repurposed medicines, which have the added value of an immediate use in clinical trials since their safety is already confirmed from their first indication. As stated by Masoudi et al. (2020) [43], “Drug repurposing is a powerful strategy in the discovery scope because of the time and cost savings”. Furthermore, it is an appropriate method for finding therapies for orphan and rare diseases.

### 3.1. Strategy 1. Antifibrinolysis: ε-Aminocaproic and Tranexamic Acids

Antifibrinolytics block the plasminogen to plasmin conversion by inhibition of its enzymatic disaggregation and consequently, stabilization of the fibrin clot. Accordingly, these drugs are expected to target the wall of the telangiectases where fibrinolysis is activated [44,45].

TA and AC are antifibrinolytic agents used for HHT epistaxis. Both may be administered topically (with the agent embedded in gauze) or systemically by oral intake (500 mg/8 h or even up to 2 g/day) or intravenous administration. AC was the first antifibrinolytic used but showed thrombosis as a side effect and is therefore not recommended in patients prone to thrombosis [46,47,48,49]. In addition, TA has shown longer half-life and higher potency (10-fold) than AC [23,24].

In addition to several case reports with successful results, a study with a total of 14 patients with low risk of thrombosis and for whom their quality of life was poor due to epistaxis, were selected to take TA (500 mg/8 h) orally. TA treatment showed a decrease of nose bleedings and increase of hemoglobin levels in all patients, almost avoiding the transfusion necessity, indicating an overall improvement and no side effects of TA treatment [24]. Although the study was not a formal clinical trial, TA was safe and effective at the doses applied. To highlight, TA administered up to 3 g/day was successful in controlling a massive and life-threatening hemorrhage in an HHT patient [23]. Moreover, some published data from in vitro experiments demonstrated on ECs that TA led to increased mRNA and protein levels of endoglin and ALK1, and improved endothelial functions as tubulogenesis and migration [24]. Thus, elevated endoglin or ALK1 expression may act concomitantly to the main antifibrinolytic action, although the proposed mechanism is only based on in vitro evidence.

Nevertheless, some concern must be taken in HHT patients with the elevated levels of the coagulation protein factor VIII (FVIII) and factor V Leiden, since some reports show an HHT-related increment of these protein levels that favor thrombotic risk in these patients. Another putative risk factor is the presence of high levels of factor V Leiden. Therefore, personalized risk-benefit considerations are demanded for HHT management [50].

Generally, clotting factors levels are not altered in HHT patients (excluding patients with altered factor V or VIII expression). However, one way of shortening the time and frequency of bleeding is by displacing the balance between coagulation and anticoagulation process toward a more quick and stable clotting when the abnormal vessels (mucosa telangiectases) break. This is enhanced by the antifibrinolytics, which prompt the clotting and delay its fibrinolysis. Moreover, telangiectases have been reported to have high fibrinolytic activity by Sabbà et al. [51].

In addition, two clinical trials were performed in HHT centers to assess the benefits of TA in HHT patients with reports published in 2014. In the French ATERO assay, TA was shown effective in a multinational center study [34], while Geisthoff et al. [32], demonstrated efficacy in a double-blind clinical trial phase III-B. However, while TA has demonstrated efficiency by systemic use, when topically used by a nasal spray, it did not significantly decrease nose bleeds when compared to placebo in a clinical trial (NOSE study) conducted by Cure HHT in 2016 [28].

### 3.2. Strategy 2. Upregulating ENG and ACVRL1

#### 3.2.1. Hormonal Therapy: Specific Estrogen Receptor Modulators (SERMs)

The incidence of epistaxis has been observed to be increased in women after they have reached menopausal age, suggesting that estrogens might play a protective role in HHT-derived bleeding in women. Post-menopausal women are also affected by osteoporosis, the imbalance in the rate of bone remodeling/resorption that predisposes elders to higher chances of bone fracture. NFκB, RANK and its ligand RANKL, as well as osteoprotegerin (OPG) play major roles in this pathogenesis [52] (Figure 2). Based on the observation that pre-menopausal HHT women had fewer epistaxis, a study developed by the Yale University’s Vascular Malformation Centre attempted to treat GI bleeding of 40 transfusion-dependent HHT patients, with a mean age of 57 years, by means of estradiol treatment. Men, to avoid estradiol feminizing effects, were also treated with ethinylestradiol/norethindrone and danazol. The results were satisfactory as most of the 40 patients showed an improvement in hemoglobin levels and needed fewer blood transfusions [53].

The use of hormones to treat HHT-induced bleedings was published later in the form of case reports, but mostly without controls [55]. The main conclusion obtained from these studies was that estrogen-progesterone administered at the doses typically used for oral contraception might reduce bleedings in symptomatic HHT women, becoming a reasonable option for fertile HHT female patients. Zacharski et al. published in 2001 a case report in which the use of tamoxifen had ceased epistaxis in the long term in a post-menopausal patient, concluding for the first time that SERM was properly used to treat epistaxis in an HHT patient [56]. According to this, tamoxifen was used in two clinical trials where it again successfully decreased epistaxis. The first of these two consisted of a placebo controlled clinical trial that included both men and women, while the second comprised a long-term monitored clinical trial in which patients were administered 20 mg tamoxifen [33,54].

In the line of using SERMs to decrease HHT-related bleedings, the safety and efficiency of raloxifene hydrochloride was tested by a Spanish HHT reference unit IDIVAL (Sierrallana/Valdecilla). Raloxifene hydrochloride shows similarities with tamoxifen, also presenting beneficial effects on bone mineralization and on prevention of cardiovascular and gynecological cancer. This study included 19 post-menopausal women, previously diagnosed with osteoporosis, and compared the amount of bleeding before and after 6 months of treatment (no placebo was included in the study). Oral intake of raloxifene (60 mg/day) showed a significant reduction in both the frequency and the amount of epistaxis after 6 months of treatment, also revealing an increase in hemoglobin levels [25]. Raloxifene has been shown to be a transcriptional activator of *ENG* and *ACVRL1/ALK1* promoters, binding to their proximal regions and subsequently increasing these genes’ transcription rate in a context of in vitro experiments on ECs [25]. As a consequence, the protein levels of endoglin and ALK1 increased, thus compensating partially the haploinsufficiency suffered by HHT patients in this study [25]. In 2010, these studies resulted in the European Medicine Agency (EMA) and Food and Drug Administration (FDA) designation of raloxifene hydrochloride as the first orphan drug to treat bleedings in HHT patients (EU/3/10/730). Bazedoxifene acetate, another SERM, significantly decreased the frequency and intensity of epistaxis, while also improving hemoglobin levels as early as one month after treatment with 20 mg/day [26]. In this case, the increase of *ENG* and *ALK1* was not only observed in experiments in vitro with ECs treated with bazedoxifene, but also, in vivo, by measuring *ENG* and *ALK1* levels in macrophages derived from patients before and after bazedoxifene treatment. Bazedoxifene was also designed as orphan drug for HHT in 2014 by the EMA (EU/3/14/1367).

Of note, estrogens and SERMs, as hormonal receptor ligands, increase the transcription of different promoters, among them, those of coagulation factor genes. Thus, *ENG* and *ACVRL1* are among the stimulated genes, but are not the only targets. In relation to this fact, especially when the treatment with SERMs may upregulate coagulation factors’ genes, blood tests should be performed periodically for HHT patients under SERM treatment in order to screen for prothrombotic markers and prevent thrombotic events [57].

Finally, phytoestrogens, compounds of plant origin with structural similarities with the natural estrogen 17β-estradiol, deserve some words in this section as natural plant estrogen related products. Among them, the isoflavone genistein and the coumestan resveratrol are the most relevant in studies related to HHT. Genistein is found in numerous plant species such as soy and red clover and resveratrol in grape skin and in dried fruits and nuts. These phytoestrogens show high affinity for estrogenic receptors. Genistein and resveratrol are involved in reducing inflammation, stimulating apoptosis and inhibiting angiogenesis [58,59], and might present therapeutic benefits in HHT patients as natural analogues to SERMs and estrogens.

#### 3.2.2. Immunosuppressor Tacrolimus (FK506)

Albiñana et al. reported the efficacy of tacrolimus (FK506) in increasing endoglin and ALK1 expression [60,61]. The reason to test this drug came from a case report of an HHT patient who was administered the immunosuppressor FK506 in low doses, in combination with Aspirin and sirolimus to avoid rejection of a liver transplant. One month after the start of this treatment, it was observed that his telangiectases (both internal and external), epistaxis and anemia had all been cured [62]. Based on this report, cultured ECs were treated with tacrolimus and an increase on the protein and mRNA expression of endoglin and ALK1 and enhancement of the TGF-β1/ALK1 signaling pathway and EC functions like tubulogenesis and cell migration were observed [60,61]. These results would explain the improvement in the above-mentioned patient, by means of a partial compensation of endoglin and ALK1 haploinsufficiency. Supporting this view, five years later, Ruiz et al. reported increased ALK1 signaling pathway in HHT patient-derived EC. In an HHT animal model, tacrolimus also inhibited VEGF signaling, decreasing hypervascularization [63].

In a more clinical context, Sommer et al. published in 2019 that low doses of FK506/Advagraf decreased bleeding in an HHT patient presenting also pulmonary arterial hypertension [64]. This case report points to low doses of tacrolimus (0.5–1.5 mg/day) as the optimal range for patients with nose or GI refractory bleeding, rather than high doses (5–10 mg/day) normally used for immunosuppression in transplants [64]. An additional report by Hosman et al., including two patients dependent on HHT transfusions due to severe bleeding, demonstrates improvement after treatment with low-dose tacrolimus [65]. Currently, and according to the HHT European Federation, around 24 HHT patients are being treated “off label” with low tacrolimus doses, prescribed by HHT reference doctors, to control epistaxis and GI bleeding.

Finally, the results regarding efficacy and safety of 0.1% tacrolimus topically applied as nasal ointment of the clinical trial named TACRO have just been published. Tacrolimus nasal ointment did not result in improvement 6 weeks after finishing treatment, but the good tolerance and the significant improvement in epistaxis duration during treatment invited the researchers for a phase 3 trial on a larger patient population and a longer treatment time, with a main outcome of epistaxis duration during treatment [42].

#### 3.2.3. N-Acetylcysteine

Based on the premise that free O_2_-radicals might cause precapillary sphincter abnormalities, resulting in epistaxis, Gussem et al. in 2009 wondered whether antioxidants like N-acetylcysteine (NAC) could neutralize those free O_2_-radicals and reduce or avoid nose bleedings. Thus, 43 HHT patients were followed-up for frequency, severity, and duration of epistaxis after a daily treatment of 600 mg NAC for 12 weeks. There was a reduction in frequency and severity of nosebleed during the day. Male patients with *ENG* mutations experienced a significant improvement. Only an improvement trend was found in women and patients with an *ALK1* mutation [66].

Based on these results, Albiñana et al. studied the in vitro effects of NAC on endoglin and ALK1 expression levels in ECs. After NAC incubation, mRNA and protein levels of endoglin increased up to 1.5–2 folds, although there were no changes on ALK1 levels [67]. These data could suggest that the improvement experienced with their symptoms in HHT1 patients might be due, in part, due to the increase of endoglin levels after NAC treatment, which could be normalizing the nasal mucosa [66].

### 3.3. Strategy 3. Antiangiogenesis

Antiangiogenic strategies on HHT act on the mucosa to decrease or normalize its abnormal excessive vasculature. Two key angiogenic pathways in ECs are those triggered by VEGF and FGF.

#### 3.3.1. Anti-VEGF and Tyrosine Kinase Inhibitors (TKI)

VEGFs specifically act on vascular EC and are a key stone in the angiogenic and lymphangiogenic process in both physiological and pathological conditions such as tumors or wound healing [68]. VEGFs play an important role in HHT since high protein levels have been reported in HHT patients [69,70,71,72].

Avastin ranks the first antiangiogenic therapeutic agent approved for advanced colorectal cancer [73]. Since then, BZ has been widely administered in other pathologies such as non-small cell lung cancer diabetic retinopathy or age-related macular degeneration [74].

The antiangiogenic properties of Avastin were successfully tested in isolated cases of HHT patients. BZ reverted the need for transplantation in a patient with HHT1 [75]. It also decreased the transfusion requirements and cardiac output in a patient with GI [76]. Thus, the French HHT Network designed a single center phase II clinical trial to address the delay in liver transplantation on HHT patients with serious liver complication [27]. BZ significantly decreased the cardiac output and reduced episodes of epistaxis. Nevertheless, symptoms did not disappear after withdrawal of the drug, making Avastin unsuitable as a surrogated alternative for orthotopic liver transplant (OLT) in HHT. These results supported the designation of BZ as orphan drug for HHT in 2014 (EU/3/14/1390).

It is very difficult to determine the optimum time to perform OLT in severe complicated liver venous malformations (VMs) in HHT. OLT is a radical cure for liver VMs and it should be the therapeutic choice in patients under the age of 65 years due to its excellent outcomes, being BZ probably a better option for patients over 65, much more susceptible to higher risk derived from surgery [77].

BZ has been also administered for very severe epistaxis. Despite of its good results decreasing epistaxis and GI after intravenous administration, it has also serious side effects. Therefore, in those patients where no other option is available, an individualized BZ treatment should be supplied; since HHT is a chronic disease [27,77,78]. This individualized re-dosing strategy will not compromise the patient’s safety or quality of life and will significantly reduce the costs [79].

BZ has also been assayed topically as nasal spray for epistaxis. The NOSE assay (under the Cure HHT support in USA, 2011–2015) and the French Ellipse clinical trial (2011–2012) ran in parallel. Unfortunately, no significant improvement could be demonstrated for the BZ vs. saline solution spray [28,29].

An alternative therapeutic approach for reducing epistaxis and GI bleeding caused by the over activated VEGF pathway is the blockade of the tyrosine-kinase activity of the VEGF receptor. As a proof of concept, the therapeutic effect of the TKI GW771806 (a pazopanib analogue), was tested in an Alk1-inducible knockout (iKO) murine HHT2 model. The oral administration of GW771806 significantly improved anemia and GI bleeding in HHT2 mice [80]. Unfortunately, its corresponding phase II human trial designed to follow-up adult HHT patients with significant epistaxis, anemia, or with transfusions could not be completed. A prospective, multi-center, open-label, dose-escalating study on pazopanib has also been published and the results show promising improvements in hemoglobin levels and epistaxis in treated patients [81].

Related to pazopanib and in a further step, nintedanib, a TKI targeting growth factor receptors involved in angiogenesis: platelet-derived growth factor receptor (PDGFR), FGFR and VEGFR, will be assayed as a therapeutic drug in a clinical trial. Nintedanib treatment in combination with rapamycin, synergizes to completely block the AVMs in HHT mice pre-clinical models [82]. This has been the main rationale to continue with nintedanib in clinical trials. In that sense, Epicure, a randomized, multicenter, phase II, double-blind placebo-controlled study promoted by the French Hospices Civils de Lyon, has started its recruitment in 2019. Epicure will test the antiangiogenic benefits of the TKI nintedanib in epistaxis. Initially, patients will be monitored for 24 weeks: 12-weeks of oral treatment plus 12-weeks of follow-up. Theoretically, nintedanib, as a non-specific/wide range TKI, should allow a reduction of epistaxis in HHT [83].

Lastly, in this section dealing with drugs interfering with the VEGF signaling pathway, thalidomide deserves special mention, since it has been used for epistaxis and GI bleeding in HHT. Its mechanism of action was published in 2010 [35] and since then, several case reports and clinical trials have been performed (Table 2 and Table 3) prior to its designation as an orphan drug (Table 4). Evidence supporting thalidomide compares favorably in cases of serious and refractory GI and nosebleeds. However, risk vs. benefit should be carefully studied, because of the reported side effects for thalidomide [78]. A derivative of thalidomide with potentially fewer side effects, pomalidomide, has been promoted for a multicenter randomized double-blind placebo controlled clinical trial currently ongoing sponsored by the NIH (Table 2).

#### 3.3.2. Non-Selective Adrenergic β-blockers

Non-selective adrenergic β-blockers of receptors β1 and β2 such as the known propranolol and timolol, have shown antiangiogenic properties related to vasoconstriction, inhibition of EC migration and proliferation and reduced VEGF expression [30,84,85,86]. Given that an excessive activation of the VEGF pathway is involved in the development of abnormal telangiectases, the properties of these non-selective adrenergic β-blockers may be useful topically. Propranolol and timolol have been used as effective therapies to treat superficial infantile hemangiomas [84,85,86] and could be considered as a potential treatment option for HHT patients.

The use of topical timolol has been described in some case reports as well as in clinical trials/studies of HHT. According to several reports, topical timolol (0.5% ophthalmic solution) clearly improved the frequency and severity of epistaxis [87,88]. Those studies were done in individual patients as well as in larger groups of HHT patients, with clear positive results in all cases. Even though no secondary adverse effects were observed in these cases, there are some contraindications for its applications which should be kept in mind when prescribing, because β-blockers can cause respiratory and cardiovascular problems. In fact, β-blockers are metabolized in the liver by CYP2D6 and a decreased expression of this enzyme, either by concomitant treatment with CYP2D6 inhibitors or by genetic variants, may lead to strong sinus bradycardia [89,90]. Timolol has also been proved to be an efficient treatment at lower doses (0.1% ophthalmic solution-timogel) in decreasing the extension and appearance of mucocutaneous telangiectases in an HHT clinical study, showing 100% and 75% improvement in HHT2 and HHT1 patients, respectively [91].

Propranolol has also been administred topically in gel formulation and systemically in tablets with very promising results, although the last method could be associated with the above-mentioned side effects, which must be considered before prescribing.

There are several studies supporting the improvement in HHT patients after propranolol topical administration. In a pilot study done with six patients, in which 1.5% propranolol gel was applied on the nasal mucosa (0.5 mL/day per nostril) the severity of epistaxis and the number of blood transfusions pre and post-administration were reduced fast and significantly [92]. As continuation of this study, the same group recently finished a double-blind placebo-controlled study to assess the efficacy and safety of topical propranolol for moderate–severe epistaxis in 24 patients with HHT [93]. The combination of sclerotherapy with polydocanol 1% and the use of propranolol cream at 0.5% prepared in a hospital pharmacy was evaluated in a cross-sectional study of 38 HHT patients. This combined therapy significantly reduced the frequency and severity of epistaxis, with an Epistaxis Severity Score (ESS) improvement of 5 points (from 6.9  ±  2.6 at the beginning to 1.9  ±  1.3 after the therapy, *p* <  0.05); and increased the quality of life of these patients (in a 5D scale, from 0.66  ±  0.27 before therapy to 0.93  ±  0.12 after the therapy *p* <  0.05) [94].

On the other hand, systemic uptake of propranolol has some side effects such as bradycardia and hypotension. Therefore, propranolol could be used in a systemic way only in HHT patients who do not have a low blood pressure. A clinical study with oral propranolol (40–120 mg/day) was done in seven hypertensive HHT patients. Among them, HHT epistaxis disappeared completely in five out of seven, while in the other two the bleeding reduction was highly significant [95].

#### 3.3.3. Antiangiogenesis by FGF Ligand Blocking

The FGF family is considered one of the largest families of polypeptide growth factors. FGFs interact with membrane tyrosine kinase receptors (FGF Receptors, FGFRs) through which they signal and exert their diverse biological functions. FGF-1 and FGF-2 were the first two pure polypeptides discovered among all FGFs and are considered one of the most potent factors promoting angiogenesis [96,97]. Later on, more factors promoting angiogenesis were described. FGFs constitutively cooperate with VEGF promoting the proliferation of EC [98] and inducing angiogenesis [99]. Consequently, FGF inhibition constitutes an alternative to the antiangiogenic effect of VEGF, becoming an interesting new therapeutic approach to treat diseases caused by uncontrolled angiogenesis. Many of the characteristics summarized previously suggest that the inhibition of the FGF signaling pathways could be an appropriate treatment to inhibit the abnormal vascularization of HHT patients [100].

FGF signaling may be inhibited in vitro and in vivo with chemical compounds. One of these products is the anion dihydroxy benzene sulfonate (2,5, DHBS), also known as dobesilate. Dobesilate is commercially available as tablets of its calcium and N-ethylene ethanamine salts (Doxium and Dicynone, respectively), or as injectable solution (Dicynone or etamsylate). Experiments done in vitro with primary cultures from healthy and HHT derived EC, show that etamsylate acts as an antiangiogenic factor, inhibiting wound healing and EC tubulogenesis [31]. A pilot clinical trial (EudraCT: 2016–003982–24, see Table 3) was performed with 12 HHT patients treated with a topical spray of etamsylate twice a day for 4 weeks. The HHT-ESS and other pertinent parameters were analyzed and registered. The significant reduction in the HHT-ESS scale (pre-treatment 4.1 vs. post-treatment 2.8), together with the lack of significant side effects, allowed the designation of topical etamsylate as a new orphan drug for epistaxis in HHT patients in 2018 (EU/3/10/18/2087) [31].

## 4. Conclusions

The currently available pharmacological treatments for HHT are summarized according to their mechanism of action on Figure 3. The drugs here included may be used on an individual level; treatment for rare diseases should follow the premises of personalized medicine, relying on the reference doctor expertise. A first-line treatment to avoid or decrease epistaxis or GI bleeding is to reinforce the speed and stability of coagulation with antifibrinolytics, preferentially TA. The only contraindication may apply to patients with risk of thrombosis. Doses vary from 1.5 g/day up to 3 g/day in episodes of severe bleeding. In countries where TA is not commercially available, AC could be used instead.

For bleedings interfering with normal life, currently the best treatment option for women in fertile age are feminine hormones used for contraception. Women in peri- or post-menopausal age can benefit from the use of SERMs. Although SERMs are primarily used to prevent or treat osteoporosis, they have shown to be effective in HHT compensating partially, the haploinsufficiency present in HHT patients by increasing the protein levels of endoglin and ALK1. Raloxifene hydrochloride was designated as orphan drug by the EMA and by the FDA. Bazedoxifene acetate was also designated as orphan drug by the EMA but it is not commercialized in United States.

Among the list of drugs that increase the expression levels of endoglin and ACVRL1/ALK1, tacrolimus was demonstrated to activate the signaling of endoglin/ALK1 in ECs. It would be the treatment of choice for immunosuppression in transplanted patients. However, recent case reports have opened the possibility of using systemic tacrolimus at low, non-immunosuppressive doses, to treat HHT bleeding. Currently, around 25 patients are using tacrolimus as an “off-label” treatment and it will be very interesting to hear about the outcome of these patients. Other treatment includes NAC which increases the endoglin RNA and protein levels and can be used as an anti-inflammatory and antioxidant drug without any side effects.

BZ is used (off-label) for antiangiogenic strategy in HHT patients with severe bleeding or symptomatic liver AVMs, in order to reduce bleedings and excessive number of abnormal mucosa vessels. TKIs (pazopanib, nintenadib, sunitinib or buparlisib) and thalidomide have also been used or are planned to be used in clinical trials [22,83,101,102] (Table 3).

Another group of antiangiogenic drugs comprises the non-selective adrenergic β-blockers like propranolol and timolol. These drugs are preferentially used topically, as creams or gels. They reduce nose bleedings by decreasing and delaying the formation of telangiectases on the mucosa. Propranolol can also be used systemically, but special attention is indicated with regards to blood pressure and heart rate.

Lastly, as emerging treatments, the orphan drug designation of etamsylate for topical treatment of epistaxis opens a new perspective following the antiangiogenic strategy.

This review has delved into the various treatment strategies of HHT and the clinical trials that support these treatments. The repurposing strategy has led to the formal approval of several orphan drugs for HHT (Table 4).

## Figures and Tables

**Figure 1 jcm-09-01766-f001:**
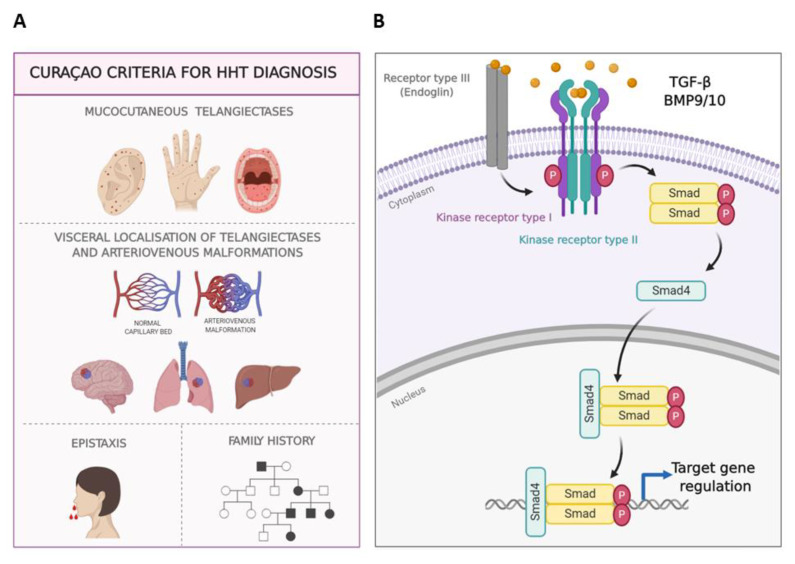
Hereditary Hemorrhagic Telangiectasia. (**A**). Clinical manifestations of HHT, Curaçao criteria. Telangiectasias in ear, hands, tongue, and lips; arteriovenous malformations in internal organs, epistaxis and family history. (**B**). TGF-β/BMP9/10 signaling pathway in endothelial cells. Once the ligand binds to its receptor complex formed by the kinase receptors I and II, and the auxiliary receptor III (endoglin), the signaling cascade leads to the phosphorylation of Smad proteins. The translocation of the Smad protein complex into the nucleus results in transcriptional regulation on target genes. Endothelial cells (EC) express two types of type I kinase Receptors: ALK1 and ALK5.

**Figure 2 jcm-09-01766-f002:**
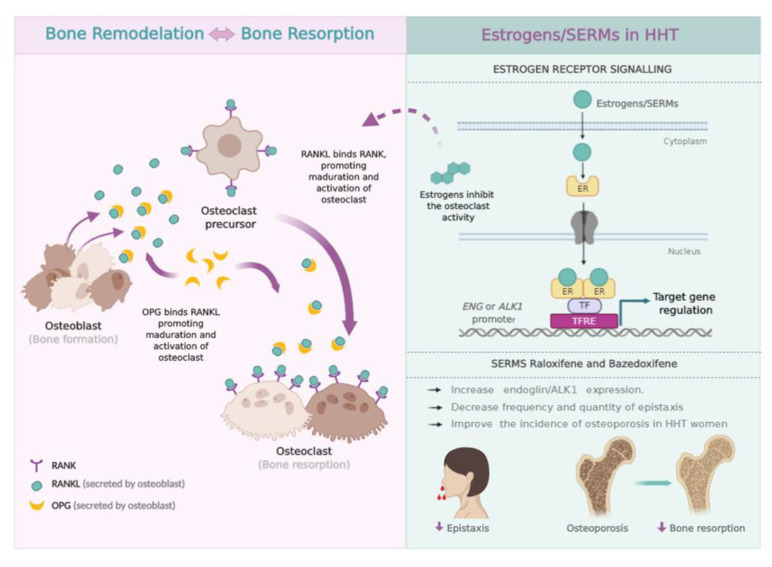
Scheme of Bone formation vs. Bone resorption (**left**). Bone remodeling is promoted by the activity of osteoclasts and osteoblasts. RANKL is a protein expressed by osteoblasts which, upon binding to its RANK receptor on the cell surface of osteoclasts and their precursors, causes RANK to stimulate and promote the adhesion of osteoclasts to bone, thus activating their function and preventing apoptosis. OPG is synthesized by the osteoblasts and acts as a decoy receptor, preventing the binding of RANKL to RANK, therefore decreasing the activity of the osteoclast and its survival [52]. For this reason, for years post-menopausal patients with diagnosed osteoporosis have been receiving estrogenic treatment, which, while correcting this imbalance in bone remodeling, also reduces the activity of the osteoclast and activates the expression of OPG. This is also capable of reducing the epistaxis of these patients [25,33,53,54]. Scheme of Estrogen Receptor (ER) signaling (**right**). Mechanism of action of ER on EC, in the case of the SERMs raloxifene and bazedoxifene. Upon ligand binding, the ER dimer interacts with different transcription factors (TF), promoting gene expression by binding to the TFRE (Transcription Factor Regulatory Element) in the promoter of its target genes. The expression of endoglin and ALK1 increases as a result of the interaction of the ER with Sp1, an essential factor for the expression of both genes. In women, SERMs promote a decrease in the frequency and amount of epistaxis, normalizing the nasal vasculature with concomitant improvement in osteoporosis [25,26].

**Figure 3 jcm-09-01766-f003:**
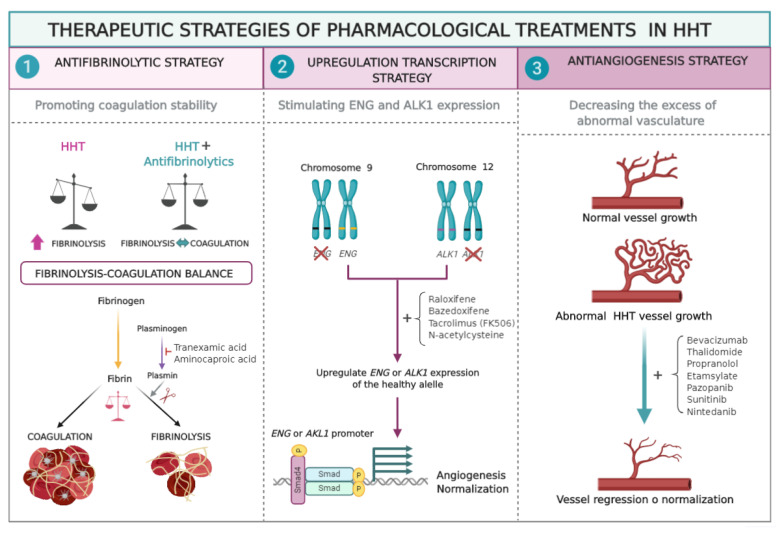
Therapeutic strategies of pharmacological treatments in HHT. (1) Antifibrinolytic strategy; prevents the conversion of plasminogen to plasmin, thus delaying the lysis of the fibrin clot, and therefore the bleeding. (2) Upregulation strategy. The drug works by increasing the expression of the *ENG* or *ALK1* genes, and thus resulting in increased amount of proteins, improving the BMP/TGF-β signaling and normalizing the formation of new vessels. (3) Antiangiogenesis strategy; makes disappear the excess of existing abnormal vasculature or normalizes it.

**Table 1 jcm-09-01766-t001:** Therapeutic strategies to decrease epistaxis in HHT.

**1**	Decrease hemorrhages stabilizing the fibrin network with antifibrinolytics
	➣ Tranexamic acid ➣ Ɛ-aminocaproicacid
**2**	Stimulate *ENG* and *ALK1* transcription to increase protein expression to partially overcome haploinsufficiency
	➣ SERMs: raloxifene, bazedoxifene ➣ Tacrolimus
**3**	Decrease the abnormal excessive vasculature of the nose mucosa through antiangiogenesis
	➣ By inactivating the VEGF signaling pathway ● Bevacizumab, propranolol, timolol, thalidomide or pazopanib ➣ By blocking the FGF-R signaling pathway ● Etamsylate

**Table 2 jcm-09-01766-t002:** Compilation of the HHT interventional clinical trials (Recruiting, Ongoing, or Unknown status).

Trial Registration #	Country	Phase	Title	Intervention	Number of Patients	Trial Design	Outcome	Start Date	Status
EudraCT 2010-020545-26	IT	2	Bevacizumab, an anti-angiogenic monoclonal antibody effective for prevention of hemorrhage in patients with HHT: possible regression of visceral arteriovenous malformations	Bevacizumab	18	Single-arm, controlled	Frequency of Epistaxis	2008	O
EudraCT 2008-006755-44	FR	2	METAFORE: Maladie de Rendu-Osler: Etude de l’Efficacité et de la tolérance du Bevacizumab utilisé pour le traitement des formes hépatiques sévères. Etude de phase II	Bevacizumab	25	Information not available	Effect on cardiac output in patients with severe liver damage	2009	O
NCT02389959	US	4	Intranasal Bevacizumab for HHT-Related Epistaxis	Bevacizumab	40	Two-arm, randomized, double-blind, placebo-controlled	Improvement in ESS	2014	R
NCT02287558	US	1	Pomalidomide in HHT and Transfusion-Dependent Vascular Ectasia: a Phase I Study	Pomalidomide	9	Single-arm, open-label	Transfusion requirement measure	2015	R
NCT04167085	US	4	NOrth American Study for the Treatment of Recurrent epIstaxis With DoxycycLine: The NOSTRIL Trial	Doxycycline	24	Two-arm, randomized, double-blind, crossover	Frequency of epistaxis	2017	A
NCT02963129	AR	3	Treatment of Nasal Staphylococcus Aureus Colonization in Patients With HHT	Mupirocin	40	Two-arm, randomized, triple-blind, placebo	Nosebleed by Sadick scale	2017	U
NCT03981562	CA	2	Vitamin D and HHT	Vit D	60	Three-arm, randomized, double-blind, placebo	Change in ESS	2018	R
EudraCT z017-003272-31	NL	2	Efficacy and safety of oral itraconazole in the reduction of epistaxis severity in HHT	Itraconazole	25	Single-arm, open-label	Change in epistaxis severity	2018	O
NCT03397004	CA	2	Doxycycline for HHT	Doxycycline Hyclate	30	Two-arm, randomized, double-blind, placebo-controlled, crossover	Reduction in epistaxis (nose bleeding) severity over 96 weeks	2018	R
NCT04113187	FR	3	Propranolol for Epistaxis in HHT a Patients	Propranolol	38	Two-arm, double-blind	Cumulative duration of epistaxis (in minutes)	2019	NR
NCT04139018	US	2	Timolol Gel for Epistaxis in HHT	Timolol Gel	30	Two-arm, double-blind, randomized controlled	ESS	2019	R
NCT03910244	US	2	Pomalidomide for the Treatment of Bleeding in HHT	Pomalidomide	159	Two-arm, placebo-controlled, double-blind	Change in ESS	2019	R
NCT03850730	US	1–2	Pazopanib for the Treatment of Epistaxis in HHT	Pazopanib	30	Single-arm, open-label	Percent change in epistaxis duration in minutes	2019	NR
NCT03850964	US	2–3	Pazopanib Effects on Bleeding in HHT	Pazopanib	45	Two-arm, double-blind, placebo controlled	Change in epistaxis duration in minutes	2019	NR
EudraCT 2019-003585-40	NL	NA	An uncontrolled, open label pilot-study assessing the efficacy in reducing bleeding severity, and the safety of oral tacrolimus in patients with HHT	Tacrolimus	20	Uncontrolled, single-arm, open-label	Change in the epistaxis and/or gastrointestinal severity	2019	O
EudraCT 2019-002593-31	FR	2	Efficacy of Nintedanib per os as a treatment for epistaxis in HHT disease. A national, randomized, multicenter phase II study EPICURE	Nintedanib	60	Two-arm, double-blind, randomized controlled, placebo	Frequency of Epistaxis	2019	O
EudraCT 2018-004179-11	NL	2	Effectiveness of Somatostatin Analogues in patients with HHT and symptomatic gastrointestinal bleeding, the SAIPAN-trial: a multicenter, randomized, open-label, parallel group, superiority trial	Somatostatin Analogues	38	Two-arm, open-label, randomized controlled	Decreasing the transfusion requirements	2019	O
ACTRN 12619001020178	AU	2	A pilot study assessing the effectiveness of oral Propranolol in preventing epistasis in patients with HHT	Propranolol	15	Single-arm, open-label, non-randomized	Change in Epistasis	2019	NR
NCT03954782	FR	2	Efficacy of Nintedanib Per os as a Treatment for Epistaxis in HHT Disease	Nintedanib	60	Two-arm, triple-blind, randomized	Epistaxis duration assessed on epistaxis grids completed by the patients	2020	O

Table 2. Compilation of the Recruiting, Ongoing, or Unknown HHT interventional clinical trials registered at the EU Clinical Trials Register (EudraCT) (https://www.clinicaltrialsregister.eu), the U.S. National Library of Medicine (NCT) (https://clinicaltrials.gov), and the Australian New Zealand Clinical Trials Registry (ACTRN) (http://www.anzctr.org.au/Default.aspx). Only interventional trials where a therapeutic drug was tested are listed. Abbreviations: Countries: AR (Argentina); AU (Australia); CA (Canada); FR (France); IT (Italy); NL (Netherlands); US (United States of America). Phase: NA (Not Applicable). Status: R (Recruiting); O (Ongoing); U (Unknown). Outcome: ESS (Epistaxis Severity Score).

**Table 3 jcm-09-01766-t003:** Compilation of the HHT interventional clinical trials (Completed or Terminated status).

Trial Registration #	Country	Phase	Title	Intervention	Number of Patients	Trial Design	Outcome (Summary of Statistically Significant Outcomes)	Start Date	Status	Link to Results
NCT00004327	US	2	Phase II Pilot Study of Octreotide, a Somatostatin Octapeptide Analog, for Gastrointestinal Hemorrhage in Hormone-Refractory HHT and Senile Ectasia	Octreotide	8	Information not available	-	1995	C	-
NCT00004654	US	3	Phase III Randomized, Placebo-Controlled, Crossover Study of Soy Protein Isolate for HHT	Soy protein	60	Randomized	-	1996	C	-
NCT01031992	GE	3	Tranexamic Acid and Epistaxis in HHT	Tranexamic acid	23	Two-arm, double-blind, controlled	No changes in hemoglobin levels. Significant reduction in ESS	2002	C	Article [32]
NCT00375622	IL	2	Anti-Estrogen Therapy for HHT A Double-Blind Placebo-Controlled Clinical Trial	Tamoxifen	25 (60) *	Double-blind, placebo-controlled	Significant reduction in ESS (frequency and severity).	2005	C	Article [33]
NCT00355108	FR	3	ATERO: A Randomized Study with Tranexamic Acid in Epistaxis of Rendu Osler Syndrome	Tranexamic acid	118 (170) *	Single-arm, double-blind, randomized, crossover	Significant decrease in the duration of epistaxis	2006	C	Article [34]
NCT00389935	US	2	Thalidomide Reduces Arteriovenous Malformation Related Gastrointestinal Bleeding	Thalidomide	14	Single-arm, open-label	-	2006	C	-
NCT00588146	US	2	Phase 2 Study of PEG-Intron in HHT	Pegylated IFN-α2B	10	Two-arm, open-label, randomized	Adverse events plus discontinuation of the study supply	2007	T	Trial file
NCT01397695	US	2	Topical Bevacizumab for the Management of Recurrent Epistaxis in Patients with HHT	Bevacizumab	20	Single-arm, open-label, non-randomized	-	2009	T	Trial file
NCT01402531	US	2	Submucosal Bevacizumab for the Management of Recurrent Epistaxis in Patients with HHT	Bevacizumab	10	Single-arm, open-label, non-randomized	-	2010	T	Trial file
EudraCT2009-018049-19	AT	2	A randomized double-blind placebo-controlled trial of intranasal submucosal bevacizumab in hereditary hemorrhagic telangiectasia	Bevacizumab	30	Two-arm, double-blind, randomizedcontrolled	-	2010	C	-
NCT01408030	US	2	North American Study of Epistaxis in HHT (NOSE)	Bevacizumab- Estriol- Tranexamic Acid	121	Four-arm, double-blind, placebo-controlled, randomized	No significant differences on epistaxis frequency	2011	C	Article [28]
NCT01408732	US	1	Office-sclerotherapy for Epistaxis Due to HHT	Sclerotherapy with sodium tetradecyl sulfate	18	Two-arm, open-label, randomized-controlled	No significant differences on severity of epistaxis	2011	C	Trial file
NCT01485224	IT	2	Efficacy of Thalidomide in the Treatment of HHT	Thalidomide	31	Single-arm, open-label, non-randomized	100% showed a complete or partial response to epistaxis	2011	C	Article [35]
NCT01314274	AT	2	Intranasal Submucosal Bevacizumab for Epistaxis in HHT	Bevacizumab	15	Two-arm, double-blind, randomized, placebo-controlled	No significant differences on epistaxis frequency	2011	C	Article [36]
NCT01507480	FR	1	The ELLIPSE Study: A Phase-1 Study Evaluating the Tolerance of Bevacizumab Nasal Spray to Treat Epistaxis in HHT	Bevacizumab	42	Single-arm, double-blind, randomized, placebo-controlled	Bevacizumab nasal spray is safe	2011	C	Article [37]
EudraCT 2011-004096-36	IT	2	Efficacy of thalidomide in the treatment of severe recurrent epistaxis in HHT	Thalidomide	31	Single-arm, non-controlled	-	2011	C	-
NCT01908543	UK	NA	Iron Deficiency and HHT	Ferrous sulphate	3	Single-arm, open-label	-	2013	T	Article [38]
NCT01752049	CA	NA	Topical Anti-angiogenic Therapy for Telangiectasia in HHT: Proof of Concept	Timolol maleate	5	Two-arm, double-blind, placebo-controlled	-	2013	C	-
EudraCT 2013-004204-19 & NCT02157987	FR	1-2	Treatment of HHT of the Nasal Mucosa by Intranasal Bevacizumab: Search for Effective Dose	Bevacizumab	30	Four-arm, double-blind, randomized, controlled	-	2014	C	-
NCT02638012	CA	NA	Prospective Pilot Study of Floseal for the Treatment of Anterior Epistaxis in Patients With HHT	Floseal	10	Two-arm, open-label, non-randomized	No statistically significant difference noted in ESS	2015	C	Article [39]
NCT02204371	US & CA	2	Evaluation of Pazopanib on Bleeding in Subjects with HHT	Pazopanib	7	Two-arm, open-label, non-randomized	No statistically significant difference noted in ESS	2015	T	Trial file
EudraCT 2015-000385-55 & NCT02484716	FR	2	Efficacy of a Timolol Nasal Spray as a Treatment for Epistaxis in Hereditary Hemorrhagic Telangiectasia (HHT)—(TEMPO)	Timolol	58	Two-arm, double-blind, randomized	No statistically significant difference noted in ESS	2015	C	Article [40]
EudraCT 2016-003982-24	ES	4-2	A phase IV-II, single-center, open, single arm treatment, low level of intervention, to assess the efficacy clinical trial and safety of intranasal administration of ethamsylate in the treatment of HHT, during for 4 weeks	Ethamsylate	12	Single-arm, open-label, non-randomized	Statistically significant difference noted in ESS	2016	C	Article [31]
EudraCT 2016-001340-19 & NCT02874326	NL	2	Octreotide in Patients With GI Bleeding Due to Rendu-Osler-Weber	Octreotide LAR	15	Single-arm, open-label	The median number of RBC units needed decreased significantly.	2016	C	Article [41]
NCT03152019	FR	2	Efficacy and Safety of a 0.1% Tacrolimus Nasal Ointment as a Treatment for Epistaxis in Hemorrhagic Hereditary Telangiectasia (HHT) (TACRO)	Tacrolimus	50	Two-arm, double-blind, randomized	No statistically significant difference noted in ESS	2017	C	Article [42]

Table 3. Compilation of the Completed or Terminated HHT interventional clinical trials registered at the EU Clinical Trials Register (EudraCT) (https://www.clinicaltrialsregister.eu), the U.S. National Library of Medicine (NCT) (https://clinicaltrials.gov). Only interventional trials where a therapeutic drug was tested are listed. Abbreviations: Countries: AT (Austria); CA (Canada); ES (Spain); FR (France); IL (Israel); IT (Italy); NL (Netherlands); UK (United Kingdom); US (United States of America). Phase: NA (Not Applicable). Status: C (Completed) and T (Terminated). Outcome: ESS (Epistaxis Severity Score). * The Study Record Detail from the website refers to the number of patients in parenthesis.

**Table 4 jcm-09-01766-t004:** Orphan drugs approved by the EMA for HHT.

Active Substance	Date	EU-Designated Number	Sponsor	Degree of Evidence Clinical Trial
Etamsylate	11/2018	EU/3/10/18/2087	CSIC, Spain	EudraCT: 2016–003982–24
Thalidomide	02/2017	EU/3/17/1845	PlumeStars, S.R.L., Italy	Several clinical trials, Table 3
Bevacizumab	12/2014	EU/3/14/1390	Dupuis-Girod, France	Several clinical trials, Table 3
Bazedoxifene	11/2014	EU/3/14/1367	CSIC, Spain	Observational study, 5 patients
Raloxifene	06/2010	EU/3/10/730	CSIC, Spain	Observational study, 19 patients

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
