# Peer review of "Review of Pharmacological Strategies with Repurposed Drugs for Hereditary Hemorrhagic Telangiectasia Related Bleeding"

_jcm, 2020, doi:10.3390/jcm9061766_

Round 1

Reviewer 1 Report

This is a comprehensive and easy to read review on therapeutic approaches to the treatment of HHT by Dr. Botella and colleagues. The authors should be congratulated. I have only a few suggestions for improvement that mainly relate to the figures and editorial details.

Figure 1b. The authors show the TGFB signaling pathway rather than the BMP pathway. If they want to show a “generic” TGF beta/BMP signaling pathway maybe they should instead use generic names such as “ligand”, “receptor kinase type II”, “receptor kinase type I” etc alternatively indicate both pathways

Figure 2. I do not see the reason that the right hand side of the Figure is included. What relevance does this have to HHT? Of more interest would be additional details on how estrogen and SERMs mechanistically affect the blood vessels in HHT.

The manuscript is well written, but there are some unusual phrases and word usage suggesting the authors’ first language is not English. I have taken the liberty of suggesting some alternative phrases or words (something I do not normally do), in order to make certain sentences clearer.

Please also use standard gene symbols (in italics: ENG, ACVRL1/ALK1, SMAD4) and protein names endoglin, ALK-1, SMAD4, where appropriate, and make tis consistent throughout the review. There are many cases where there are no spaces between words, especially for the in text citations. Please make this consistent throughout.  When used within a sentence, drug name do not normally begin with an upper case letter, unless this is a trademark, please correct throughout.

Author Response

A word document with the answers to the reviewer 1 has been uploaded

Reviewer 2 Report

This review summarises the different pharmaceutical approaches being used to treat HHT patients and the basis of their use. This is a timely review and is more comprehensive than a recent related review (Robert et al 2020). However, the review is extremely wordy and the English grammar is quite poor. It requires detailed correction by a native English speaker. I have a few additional comments.

It was not clear conceptually why anti-fibrinolytics are beneficial when clotting is normal in HHT. This could be explained more clearly.

Table 2 is difficult to interpret. It would be helpful to split table 2 into two tables, one for completed trials and one for ongoing and planned trials. For completed trials the authors could add the number of patients, and a key to trial design (ie whether or not a double blind placebo controlled trial), as well as a summary of statistically significant outcomes. These are more important details than the formal name of the trial and would help to communicate to the reader the level of confidence in the findings. For ongoing and planned trials the design and expected patient number would be useful additions, whilst the outcomes being measured are sufficient as already provided. This change would aid a better understanding of the status of trials in HHT patients.

Adverse side effects were mentioned for several of the drugs, but the seriousness of the side effects was not clear. In particular, whether or not they were sufficiently severe to halt use of the drug in question in current or future trials in HHT patients.

There are a number of issues with the signalling pathway in Figure 1. First the pathway uses TGFbeta as a ligand whereas HHT is currently thought to be a disease of BMP9/10 signalling.  The signalling pathway also refers to TGFBR1, which is ALK5, rather than ALK1, as the signalling receptor, and the type II receptor may be either BMPRII or ACTRIIB. In addition, the signalling pathway includes phosphorylation of SMAD4. This phosphorylation does occur downstream of ERK, STK11 and p38 pathways in some cases inhibiting signalling. This is a complexity of control that is not required to explain in the figure so it is easily rectified by removing the phosphorylation from SMAD4.

The detailed consideration of bone formation seemed out of place as it deflected focus on the HHT disease, where abnormal bone development and healing are not a feature of disease.

Author Response

The point by point answers are uploaded from an attached word file
